# Assessment of *in vitro* activities of novel modified antimicrobial peptides against clarithromycin resistant *Mycobacterium abscessus*

**Phantitra Sudadech**[1,2], **Sittiruk Roytrakul**[3], **Orawee Kaewprasert**[1,2], **Auttawit Sirichoat**[1,2], **Ploenchan Chetchotisakd**[2,4], **Sakawrat Kanthawong**[1,2], **Kiatichai Faksri**[1,2]*

1 Department of Microbiology, Faculty of Medicine, Khon Kaen University, Khon Kaen, Thailand, 2 Research and Diagnostic Center for Emerging Infectious Diseases (RCEID), Khon Kaen University, Khon Kaen, Thailand, 3 Genome Institute, National Center for Genetic Engineering and Biotechnology, National Science and Technology Development Agency (NSTDA), Pathum Thani, Thailand, 4 Department of Medicine, Faculty of Medicine, Khon Kaen University, Khon Kaen, Thailand

* kiatichai@kku.ac.th

**Data Availability Statement:** All relevant data are within the manuscript and its Supporting Information files.

## Abstract

*Mycobacterium abscessus* (*Mab*) is one of the most drug resistant bacteria with a high treatment failure rate. Antimicrobial peptides (AMPs) are alternative therapeutic agents against this infection. This study was aimed to assess the *in vitro* activities of thirteen AMPs (S5, S52, S6, S61, S62, S63, KLK, KLK1, KLK2, Pug-1, Pug-2, Pug-3 and Pug-4) that have never been investigated against drug resistant *Mab* isolates. Only four novel modified AMPs (S61, S62, S63 and KLK1) provided the lowest minimum inhibitory concentration (MIC) values ranging from 200–400 µg/ml against the *Mab* ATCC19977 strain. These four potential AMPs were further tested with 16 clinical isolates of clarithromycin resistant *Mab*. The majority of the tested strains (10/16 isolates, 62.5%) showed ~99% kill by all four AMPs within 24 hours with an MIC <50 µg/ml. Only two isolates (12.5%) with acquired clarithromycin resistance, however, exhibited values <50 µg/ml of four potential AMPs, S61, S62, S63 and KLK1 after 3-days-incubation. At the MICs level, S63 showed the lowest toxicity with 1.50% hemolysis and 100% PBMC viability whereas KLK1 showed the highest hemolysis (10.21%) and lowest PBMC viability (93.52%). S61, S62 and S63 were further tested with clarithromycin-AMP interaction assays and found that 5/10 (50%) of selected isolates exhibited a synergistic interaction with 0.02–0.41 FICI values. This present study demonstrated the potential application of novel AMPs as an adjunctive treatment with clarithromycin against drug resistant *Mab* infection.

## Introduction

*Mycobacterium abscessus* (*Mab*) is one of the species of non-tuberculous mycobacteria (NTM) that can cause various human diseases [1]. This pathogen is one of the most resistant bacteria

**Funding:** This study was supported by the Invitation Research, Faculty of Medicine, Khon Kaen University (Grant number: IN62307) and National Research Council of Thailand (Grant number: NRC MHESI 483/2563). The funders had no role in study design, data collection and analysis, decision to publish, or preparation of the manuscript.

**Competing interests:** The authors have declared that no competing interests exist.

to the current antibiotics [2]. *Mab* strains could be further divided into three closely related taxa, i.e., subspecies *abscessus*, subspecies *massiliense* and subspecies *bolletii* [3]. In the past 20 years, the incidence of *Mab* infection has increased [4].

According to the ATS/IDSA guidelines, macrolide antibiotics, especially clarithromycin combined with intravenous amikacin and cefoxitin or imipenem were the recommended treatments of choice for *Mab* infection [5]. The duration of treatment for *Mab* infection depends on the clinical syndrome and lasts from 4 weeks to 12 months [6, 7]. The high antibiotic resistance and treatment failure rate of *Mab* infection, is, however, still a great obstacle [2]. In the last decade, clarithromycin resistant *Mab* has increased [8]. There is also, a situation in that the pharmaceutical industry has reduced the development of new antibiotics due to the cost-effectiveness and rapid development of drug resistance to novel antibiotics [9]. Alternative treatment approaches and/or improvement of the current treatment of drug resistant *Mab* infections are urgently needed.

Antimicrobial peptides (AMPs) are one of the alternative treatments against drug resistant *Mab* that have broad-spectrum antimicrobial activities [10, 11]. Several research teams have reported AMPs activity against *Mycobacterium tuberculosis* [12–29] and other NTMs such as *Mycobacterium avium* [19, 27, 30, 31], *Mycobacterium smegmatis* [20, 27, 32], *Mycobacterium vaccae* [33], *Mycobacterium bovis* [34] and *Mycobacterium marinum* [35]. Previously, there have been few studies that have investigated the activities of AMPs against *Mab*. NDBP-5.5 at 200 μM showed a minimal bactericidal concentration (MBC) against three clinical isolates of *Mab* subsp. *massiliense* with low hemolytic toxicity [36]. Polydim-I treatment of macrophages infected with different *Mab* subsp. *massiliense* strains reduced the bacterial load by 40 to 50% [37]. ToAP 2 at 200 μM MBC inhibited the replication of four *Mab* subsp. *massiliense* strains [38]. These studies, however, did not investigate AMPs among the subspecies of *Mab* or made comparisons between strains with inducible or acquired resistance. Furthermore, no study investigated the antimicrobial activity of AMPs against *Mab* when combined with clarithromycin.

In this study, it was aimed to evaluate the AMPs that demonstrated antimicrobial activities against drug resistant bacteria as alternative therapeutic agents against *Mab*. The novel AMPs based on modifications by truncation of amino acid sequences of AMPs (S5, S6 and KLK) were also tested against clarithromycin resistant *Mab*. This study determined the activities of these AMPs based on their toxic effects and combination effects between these peptides and clarithromycin.

## Materials and methods

### Culture, identification and DNA extraction from *Mab* isolates

Sixteen clinical isolates of *Mab* were obtained from patients at the Clinical Laboratory Unit, Srinagarind Hospital, Khon Kaen University, Khon Kaen, Thailand between 2012 to 2016 (**S1 Table**). All specimens were fully anonymized before they were accessed. The species identification of *Mab* was performed according to protocols published previously [39]. The isolates were preserved in Middlebrook 7H9 (Difco, Detroit, MI, USA) supplemented with oleic acid-albumin dextrose-catalase (OADC) (BBL, Becton Dickinson, USA) plus 20% glycerol at -20°C. All *Mab* isolates were re-subcultured on Löwenstein–Jensen (LJ) medium at 37°C for 3–5 days. Genomic DNA of *Mab* isolates were extracted from loops full of colonies using the cetyl-trimethyl-ammonium bromide-sodium chloride (CTAB) method [40]. Subspecies of *Mab* were identified based on multilocus sequence typing (MLST) as in a previous study [41]. Informed consent was not required for this study. All specimens including isolates and blood samples were obtained from routine practice in which patient information was deidentified.

The study protocol was approved by the Khon Kaen University Ethics Committee for Human Research (HE611496).

## *In vitro* susceptibility testing of clarithromycin

Drug susceptibility testing (DST) was performed according to the Clinical and Laboratory Standards Institute (CLSI) guidelines M24-A2 [42] using the broth microdilution method to determine the minimum inhibitory concentration (MIC). Two-fold serial dilutions of clarithromycin and amikacin (Sigma-Aldrich, Oakville, ON, Canada) were prepared in a 96-well plate with Mueller-Hinton broth ranging from 0.5 to 1,024 μg/ml. Colonies were grown at an adjusted cell density to a 0.5 McFarland standard and further diluted to $5 \times 10^5$ CFU/ml. This inoculum was added to each well of the 96-well plates containing different concentrations of clarithromycin. These were then incubated at 37°C for 3, 5 and 14 days. The MIC was defined as the concentration in which no visible growth was observed. The results were interpreted according to the guidelines of CLSI. Inducible resistance was inferred by changes in MIC values from being susceptible at day 3 to resistant at day 14. Strains with a resistance status since day 3 were regarded as demonstrating acquired resistance.

## AMPs used in this study

Thirteen AMPs including, S5, S52, S6, S61, S62, S63, KLK, KLK1, KLK2, Pug-1, Pug-2, Pug-3 and Pug-4 (Table 1) were provided from the National Center for Genetic Engineering and Biotechnology (BIOTEC), Thailand. These AMPs were randomly selected based on potential antimicrobial activity against drug resistant bacteria from the literature and/or never having been tested against drug resistant *Mab*. Three parent AMPs (S5, S6 and KLK) were randomly

**Table 1. Characteristics and properties of antimicrobial peptides used in this study.**

| AMP codes | Sources | Molecular weights (Da) | Amino acid sequences | Net charges | Hydrophobicity (%) | pI | MIC values (μg/ml)[e] |
|---|---|---|---|---|---|---|---|
| S5 | *Buthus martensii* Kasch [a] | 1,448.79 | FIGAIARLLSKIF | 2 | 56.23 | 11.6 | >400 |
| S52 | *Buthus martensii* Kasch [a, #] | 1,188.46 | FIGAIARLLSK | 2 | 66.67 | 11.6 | >400 |
| S6 | Bovine myeloid cells [b] | 2,059.46 | GGLRSLGRKILRAWKKYG | 6 | 33.33 | 11.91 | >400 |
| S61 | Bovine myeloid cells [b,#] | 2,002.41 | GGLRSLGRKILRAWKKY | 6 | 35.29 | 11.91 | 200 |
| S62 | Bovine myeloid cells [b,#] | 1,839.24 | GGLRSLGRKILRAWKK | 6 | 37.5 | 12.44 | 200 |
| S63 | Bovine myeloid cells [b,#] | 1,711.07 | GGLRSLGRKILRAWK | 5 | 40 | 12.43 | 200 |
| KLK | *Sarcophaga peregrina* [c] | 1,322.81 | KLKLLLLLKLK | 4 | 63.64 | 11.15 | >400 |
| KLK1 | *Sarcophaga peregrina* [c,#] | 1,194.66 | KLKLLLLLKL | 3 | 70 | 10.98 | 400 |
| KLK2 | *Sarcophaga peregrina* [c,#] | 1,081.50 | KLKLLLLLK | 3 | 66.67 | 10.98 | >400 |
| Pug-1 | *Punica granatum* [d] | 1,553.84 | LLKLFFPFLETGE | -1 | 61.54 | 4.15 | >400 |
| Pug-2 | *Punica granatum* [d] | 587.67 | GAVGSVV | 0 | 57.14 | 3.65 | >400 |
| Pug-3 | *Punica granatum* [d] | 452.5 | LGTY | 0 | 25 | 3.61 | >400 |
| Pug-4 | *Punica granatum* [d] | 922.08 | FPSFLVGR | 1 | 62.5 | 10.59 | >400 |

**Note:** AMP, antimicrobial peptide; Da, daltons; pI, isoelectric points; MIC, minimal inhibitory concentration.

[a] *Buthus martensii* Kasch (scorpion venom) [44].

[b] bovine myeloid cells [45].

[c] *Sarcophaga peregrina* (flesh fly) [46].

[d] *Punica granatum* (Pomegranate peel) [47].

[e] *In vitro* screening antimicrobial activities of 13 AMPs against *M. abscessus* ATCC19977 strain. MIC values were measured in duplicate in two independent experiments.

[#] Novel modified AMPs by truncation of amino acid residues from its parent AMP from the current study.

modified by truncation of amino acid residues from the parent AMP (**Table 1**). AMPs were synthesized by China Peptides Co., Ltd. (Shanghai, China) or GenScript (Piscataway, USA). The purity of AMPs was >90%. Their molecular weights, net charges, percent hydrophobicity and isoelectric points (pIs) were calculated using APD3 the Antimicrobial Peptide Calculator and Predictor [43].

## Antimicrobial screening assay of AMPs

The antimicrobial assay was screened with the *Mab* ATCC19977 strain to determine the MIC values of 13 AMPs by the broth microdilution method as described above. Briefly, serial dilutions of the AMPs were prepared with potassium phosphate buffer (PPB) from the concentration range of 3.125 to 400 μg/ml and then 50 μl of each dilution were added to each 96-well plate. Colonies of the isolate were suspended and then further diluted in Mueller-Hinton broth to obtain a final concentration of $1 \times 10^3$ CFU/ml (optimized according to the available AMP stock concentration). Fifty microliters of this inoculum were added to each well of the plates. The plates were incubated for 3 days at 37°C. Plain media and bacterial suspensions without AMPs were used as negative and positive controls. The MIC values were read and recorded. All assays were performed in duplicate and two independent experiments.

## 24-hour bactericidal activity assays and *in vitro* antimicrobial susceptibility of potential AMPs against clinical isolates of *Mab*

Potential AMPs from the screening assays were subjected to determine the 24-hour bactericidal activity assays (corresponding to the common time of administration) with sixteen clinical *Mab* isolates using the protocol as described above. To observe the early antimicrobial activities, the plates were incubated for 24 h at 37°C. The samples from each well were further diluted in 0.05% Tween 80 and inoculated on Mueller-Hinton agar. After incubation for 3 days at 37°C, the colony-forming units (CFUs) were counted and calculated for the percentage of killing using the following formula: % killing = 1 - (CFU sample/ CFU control) ×100.

For the *in vitro* antimicrobial susceptibility, the same protocol as described above was used. The culture plates were further incubated up to 3 days at 37°C. The MIC values were read and recorded.

## Toxicity assays of the potential AMPs in human blood cells

For the hemolytic toxicity test, red blood cells (RBCs) from fresh blood samples of healthy volunteers were obtained by centrifugation at 1,000 rpm for 5 min. The cells were washed three times with sterile phosphate buffered saline (PBS) and adjusted into $1 \times 10^8$ cells/ml. AMPs were diluted with PPB in ranging from 6.25 to 400 μg/ml. Fifty microliters of the RBC suspensions and 50 μl of AMP solutions were added to 1.5 ml sterile microtubes, incubated for 1 h at 37°C and then centrifuged at 1,000 rpm for 5 min. The supernatants were transferred to new 96-well microtiter plates for measurement of the absorbance at 540 nm using a microplate reader. RBC suspensions treated with 2% (v/v) Triton X-100 and PBS solution were used as positive and negative controls. The percentages of hemolysis were calculated using the following formula: % of RBC lysis = 100 × [(Test—PBS) / (Positive control—PBS)]. All assays were performed in duplicate and two independent experiments.

For the toxicity of the potential AMPs to human peripheral blood mononuclear cells (PBMCs), the trypan blue exclusion assay was used. Fresh human PBMCs were prepared from blood samples of healthy volunteers using the Ficoll density gradient technique. The cells were centrifuged at 1,800 rpm for 20 min at 20°C and washed three times with PBS and adjusted with RPMI-1640 medium into $6.25 \times 10^5$ cells/ml. Fifty microliters of PBMC suspension and

50 µl of AMP solutions (6.25–400 µg/ml) were added to the 96-well microtiter plates and incubated for 1 h at 37°C. Then, 20 µl of the samples were mixed with 20 µl of 0.4% (w/v) trypan blue solution (0.81% NaCl and 0.06% (w/v) dibasic potassium phosphate) in microtubes and incubated for 3 min at room temperature. The PBMC suspensions treated with PBS were used as negative controls. PBMCs were counted using a dual-chamber hemocytometer under a light microscope. Viable and non-viable cells were counted under a microscope and the percentage of viable cells was calculated using the following formula: % of viable cells = [1.00 –(Number of viable cells / Number of total cells)] × 100. All assays were performed in duplicate with two independent experiments.

## Whole genome sequencing of the tested *Mab* strains

The total genomic DNA belonging to sixteen *Mab* strains was constructed with a 350-bp insert DNA library, and 150-bp paired-end reads sequenced using a Genome Sequencer Illumina HiSeq sequencing at Novogene Company Limited, Hong Kong. The quality of raw sequences was checked using the FastQC version 0.11.7 [48]. Trimmomatic (v0.36) software [49] was used to remove low-quality reads (leading:3, trailing:3, sliding window:4:15 and minlen:75). High-quality paired-end reads were then mapped to *M. abscessus* ATCC19977 reference genome (GenBank accession number CU458896.1) using BWA-mem (v.0.7.17) [50]. For converting SAM to BAM format, sorting and indexing the bam files, SAMtools v0.1.19 algorithm was used [51]. GATK version 4.0.5.2 [52] was used for realignment, generating coverage statistics and mapping details. Both GATK and SAMtools were used for variant calling and filtering, including single nucleotide polymorphisms (SNPs) and small indels. The analysis parameters (Q30, C40, QSNP30, d20% (60X) and ≥80% frequency of the main variant) were used to generate high-confidence SNPs. The WGS-based phylogeny of 16 clinical *Mab* isolates were analyzed based on the maximum likelihood (ML) method using MEGA-7 [53] with the general time-reversible (GTR) and gamma model with 1,000 bootstrap replicates. Visualization of the phylogenetic tree was performed using iTOL (https://itol.embl.de/). Raw sequences were deposited in the NCBI under the BioProject accession number PRJNA523980.

## Interaction and synergistic assays between potential AMPs and clarithromycin

The 2D-broth microdilution checkerboard technique was used [54] to determine the interaction and synergistic effects between potential AMPs and clarithromycin. Baseline MIC values of each AMP and clarithromycin from each clinical isolate were adopted from the experiments above. Briefly, seven concentrations of AMPs and clarithromycin were serially diluted from 1 to 64-fold of the baseline MIC. The combinations among AMP and clarithromycin concentrations were added in 96-well microtiter plates. *Mab* suspensions at a $5\times10^5$ CFU/ml final concentration were added to each well and incubated for 3 days and 14 days at 37°C. MIC values were defined when the percent killing of CFUs of more than 90% were compared to the media controls without AMPs or drugs. The fraction inhibitory concentration index (FICI) was calculated using the following formula: FICI = [C($MIC^A$)/$MIC^A$] + [C($MIC^B$)/$MIC^B$]. Notably, C($MIC^A$) = the MIC of compound A in combination, $MIC^A$ = the MIC of the compound A alone, C($MIC^B$) = the MIC of compound B in combination and $MIC^B$ = the MIC of the compound B alone. For interpretation, FICI ≤ 0.5 was interpreted as synergism, FICI >0.5–1.0 was interpreted as additive, FICI >1.0–4.0 was interpreted as indifferent and FICI >4.0 was interpreted as antagonism [54, 55].

## Statistical analysis

Descriptive statistics were used to describe the results in this study. One-way ANOVA, followed by the Tukey test was used for the variances among groups of the toxicity assays (duplicate with two independent experiments). *P*-values <0.05 were considered statistically significant. All statistical analyses were performed using SPSS version 19.0 (IBM, Armonk, NY, USA).

## Results

### Screening antimicrobial activities of 13 AMPs against *Mab* ATCC19977

In the results of screening antimicrobial activity, only four AMPs had MIC < 400 μg/ml. S61, S62, and S63 had an MIC of 200 μg/ml and KLK1 was at 400 μg/ml (**Table 1**). Therefore, S61, S62, S63, and KLK1 were recognized as potential AMPs and selected for further investigation.

### 24-hour bactericidal activity of potential AMPs against clinical isolates of *Mab*

The results of 24-hour bactericidal activity assays of four AMPs (S61, S62, S63, and KLK1) varied among sixteen clinical *Mab* isolates (**Fig 1** and **S1 Table**). In a majority of the tested strains (10/16 isolates, 62.5%), an ~99% were killed by all four AMPs within 24 h with an MIC

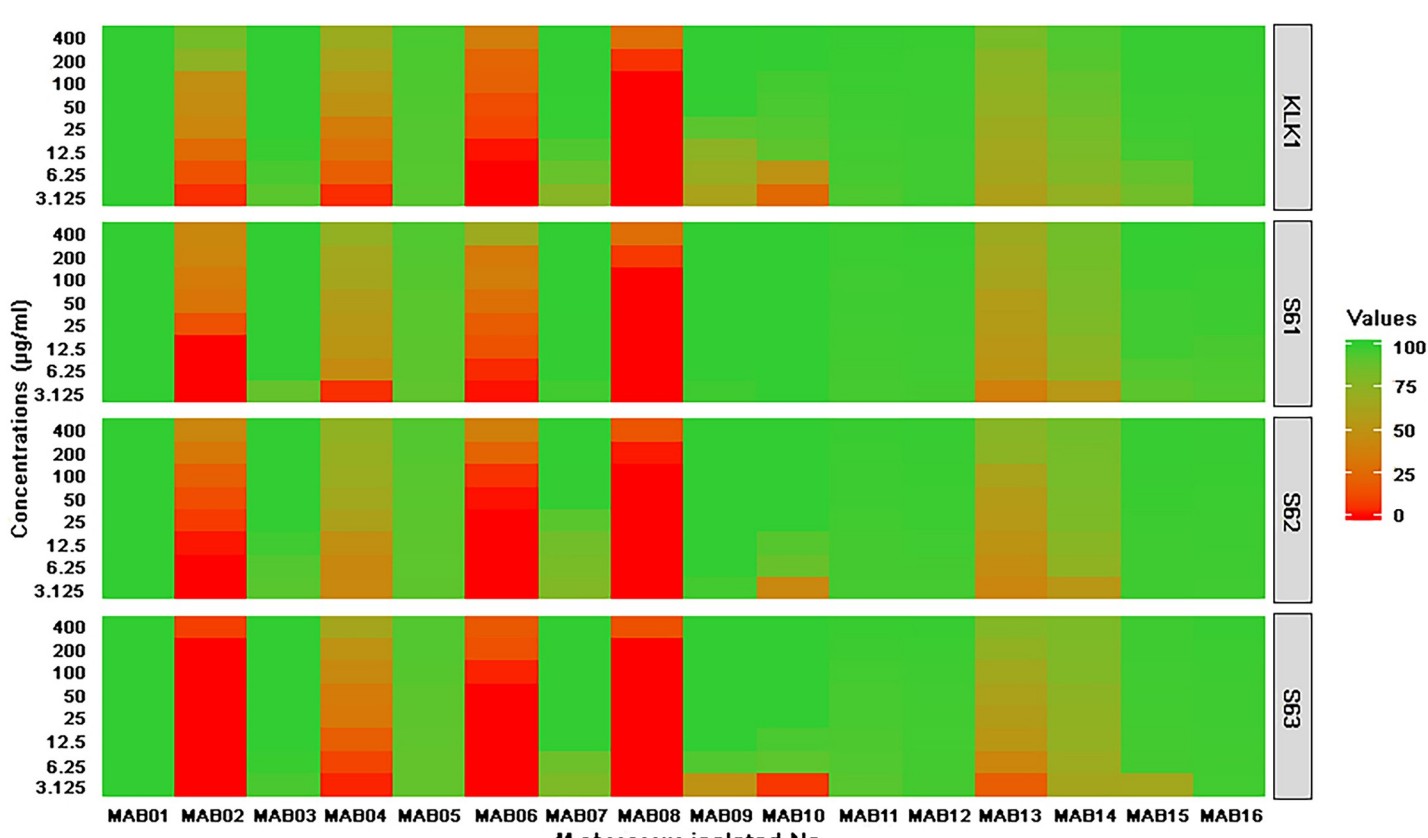

**Fig 1. The 24-hour bactericidal activities of four AMPs against sixteen clinical isolates of *M. abscessus*.** The heat map demonstrates the percentages of killing of four AMPs against each of the *M. abscessus* isolates. The green color represents high bactericidal activity (100% killing score) and the red indicates low bactericidal activity (0% killing score).

**Table 2. Characteristics and *in vitro* antibacterial activities of four potential AMPs against 16 clinical isolates of *M. abscessus*.**

| Isolates | Organism | Subspecies[a] | Colony morphology | DST profile, MIC value (µg/mL)[d] | | | | AMK | Antimicrobial peptides against *M. abscessus* isolates, MIC value (µg/mL) | | | |
|---|---|---|---|---|---|---|---|---|---|---|---|---|
| | | | | CLA | | | | | | | | |
| | | | | Day 3 | Day 5 | Day 14 | Type of resistance | Day 5 | S61 | S62 | S63 | KLK1 |
| MAB01 | *M. abscessus* | *abscessus* | Smooth | 1,024 (R) | 1,024 (R) | 1,024 (R) | Acquired | 64 (R) | 6.25 | 12.5 | 6.25 | 25 |
| MAB02 | *M. abscessus* | *abscessus* | Mixed | 1 (S) | 8 (R) | 64 (R) | Inducible | 8 (S) | >400 | >400 | >400 | >400 |
| MAB03 | *M. abscessus* | *abscessus* | Smooth | 8 (R) | 8 (R) | 8 (R) | Acquired | 4 (S) | 6.25 | 12.5 | 12.5 | 50 |
| MAB04 | *M. abscessus* | *abscessus* | Mixed | 2 (S) | 16 (R) | 64 (R) | Inducible | 8 (S) | >400 | >400 | >400 | >400 |
| MAB05 | *M. abscessus* | *abscessus* | Smooth | 4 (I) | 8 (R) | 16 (R) | Inducible | 32 (I) | >400 | >400 | >400 | >400 |
| MAB06 | *M. abscessus* | *abscessus* | Rough | 0.5 (S) | 2 (S) | 32 (R) | Inducible | 8 (S) | >400 | >400 | >400 | >400 |
| MAB07 | *M. abscessus* | *abscessus* | Smooth | 4 (I) | 16 (R) | 16 (R) | Inducible | 8 (S) | >400 | >400 | >400 | >400 |
| MAB08 | *M. abscessus* | *abscessus* | Rough | 1 (S) | 2 (S) | 8 (R) | Inducible | 8 (S) | >400 | >400 | >400 | >400 |
| MAB09 | *M. abscessus* | *abscessus* | Mixed | 0.25 (S) | 8 (R) | 8 (R) | Inducible | 8 (S) | >400 | >400 | >400 | >400 |
| MAB10 | *M. abscessus* | *abscessus* | Smooth | 1,024 (R) | 1,024 (R) | 1,024 (R) | Acquired | 64 (R) | >400 | >400 | >400 | >400 |
| MAB11 | *M. abscessus* | *abscessus* | Mixed | 0.5 (S) | 16 (R) | 256 (R) | Inducible | 16 (S) | >400 | >400 | >400 | >400 |
| MAB12 | *M. abscessus* | *massiliense* | Rough | 1,024 (R) | 1,024 (R) | 1,024 (R) | Acquired | 8 (S) | >400 | >400 | >400 | >400 |
| MAB13 | *M. abscessus* | *massiliense* | Rough | 8 (R) | 32 (R) | 32 (R) | Acquired | 8 (S) | >400 | >400 | >400 | >400 |
| MAB14 | *M. abscessus* | *massiliense* | Rough | 512 (R) | 512 (R) | 512 (R) | Acquired | 64 (R) | >400 | >400 | >400 | >400 |
| MAB15 | *M. abscessus* | *massiliense* | Smooth | 4 (I) | 4 (I) | 4 (I) | Intermediate | 8 (S) | >400 | >400 | >400 | >400 |
| MAB16 | *M. abscessus* | *massiliense* | Mixed | 0.2 (S) | 2 (S) | 2 (S) | Susceptible | 32 (I) | >400 | >400 | >400 | >400 |

**Note:** AMP, antimicrobial peptide; MLST, multilocus sequence typing; CLA, Clarithromycin; AMK, Amikacin; DST, Drug susceptibility testing; S, susceptible; I, intermediate; R, resistant.

[a] Subspecies of *M. abscessus* were identified based on MLST as in a previous study [41].

[b] The DST was performed following the method that is described above and types of CLA resistance were interpreted based on *in vitro* MIC results.

<50 µg/ml (less than 24.97 µM of S61, 27.19 µM of S62, 29.22 µM of S63 and 41.85 µM of KLK1). At the MIC levels, only two isolates (MAB01 and MAB03, 12.5%) with acquired clarithromycin resistance were 100% killed by all four AMPs within 24 h. The remaining (6/16 isolates, 37.5%) were resistant to the highest concentrations (400 µg/ml) of four potential AMPs. Similar to *Mab* ATCC19977, the patterns of susceptibility of each isolate against four AMPs were consistent. S61 had the best 24-hour bactericidal activity against sixteen isolates (Fig 1).

## *In vitro* susceptibility testing of potential peptides

Only two *Mab* isolates (MAB01 and MAB03, 12.5%) exhibited MIC values <50 µg/ml of four potential AMPs (3.13 µM of S61, 6.80 µM of S62, 3.65–7.30 µM of S63 and 20.93–41.85 µM of KLK1) after 3 days according to the incubation time of standard drug susceptibility testing (Table 2). Fourteen of sixteen or 87 percent of the isolates had values of >400 µg/ml after 3 days of incubation with all potential AMPs.

## Toxicity of potential AMPs to human RBCs and PBMCs

Variations of hemolytic activity ranged from 0.18 to 12.15% of each AMP at the MIC levels as shown in Fig 2A and S2 Table. At the MIC levels, S63 showed the lowest hemolysis (0.53±0.75 to 3.52±4.98%), whereas KLK1 showed the highest hemolysis (8.27±1.74 to 12.15±0.25%). Similarly, the variations of PBMC toxicity were varied among AMPs. At MIC levels, S63 showed the lowest PBMC toxicity (103.67±6.60 to 101.78±5.46% of viable cells) whereas it showed the highest PBMC toxicity (92.70±4.41 to 94.34±2.80% of viable cells) (Fig 2B). Due to

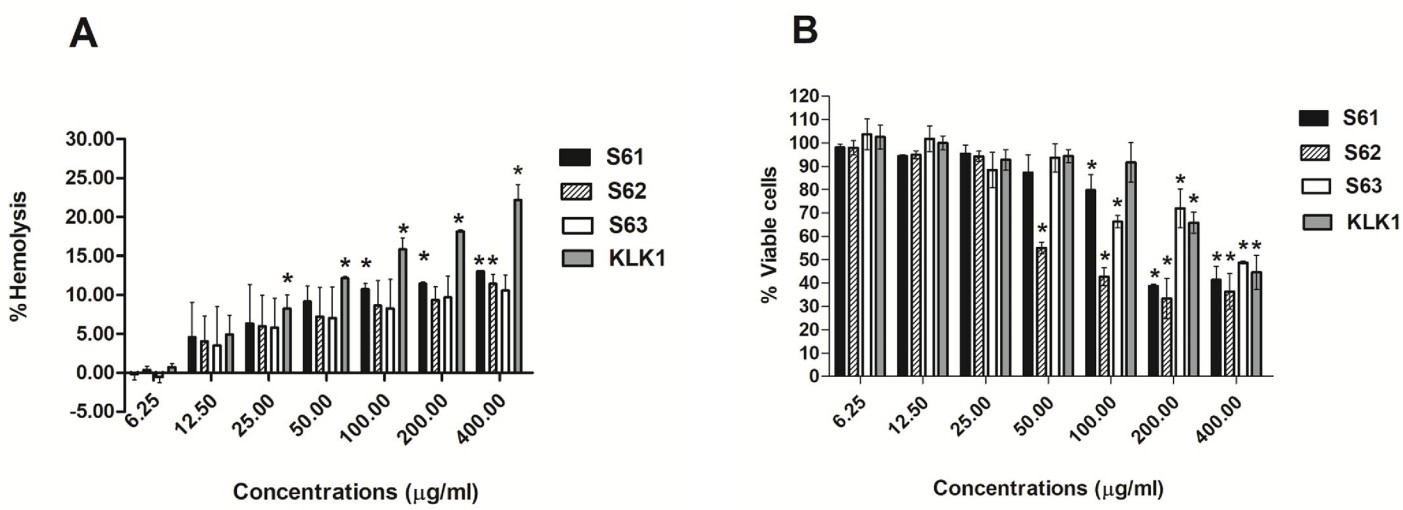

**Fig 2. Toxicity testing of 4 potential AMPs to human red blood cells (RBCs) and peripheral blood mononuclear cells (PBMCs).** Percent hemolysis of human RBCs after treatment with various concentrations of four AMPs for 1 h. (A). Percent PBMCs viability after treated with various concentrations of four AMPs for 1 h. (B). The data exhibited mean ± S.D. of duplicates from two independent experiments. One-way ANOVA followed by Tukey's test was used to determine significant differences (*$P$ <0.05). MIC of S61, S62, S63 and KLK1 were 6.25, 12.5, 9.38 and 37.5 μg/ml.

high toxicity to both human RBCs and PBMCs, KLK1 was excluded from the AMP-clarithromycin integration assay.

## Synergistic effect between the potential AMPs and clarithromycin

Three AMPs (S61, S62 and S63) were selected for the AMP-clarithromycin integration assay tested against ten clarithromycin resistant isolates which represent a clade of the whole. It was found that half of the tested isolates (5/10 isolates, 50%) exhibited synergistic interactions with 0.02–0.41 FICI values (**Table 3**). Both S61 and S62 showed the highest synergistic effects with clarithromycin. The remaining provided additive and indifferent interactions with 0.52–1.04 FICI values. The synergistic effects between AMP and clarithromycin were found in both inducible and acquired clarithromycin strains (**Fig 3**). No associations between the phylogeny or types of clarithromycin resistance (acquired and inducible resistance) and the AMP-clarithromycin synergistic effect were found (**Fig 3**).

## Discussion

In this study, the AMPs derived from *Buthus martensii* Kasch (scorpion venom), bovine myeloid cells, *Sarcophaga peregrina* (flesh fly) and *Punica granatum* (pomegranate peel) were investigated. S5 (from scorpion venom) inhibited and disrupted *Pseudomonas aeruginosa* biofilms [44] and had antimicrobial activities against *Neisseria gonorrhoeae* [56] and carbapenem resistance in *Enterobacteriaceae* (CRE) [45]. S6 (bovine myeloid) had antimicrobial activities against methicillin-resistant *Staphylococcus aureus* (MRSA) [57] and CRE [45]. Anti-inflammatory activity of KLK (flesh fly) was demonstrated [46]. Recently, the antibiofilm effect of novel AMPs extracted from pomegranate (*Punica granatum*) on *Streptococcus mutans* adhesion was reported [47]. These AMPs were in the candidate pool that this study planned to test for potential antimicrobial activity against drug resistant bacteria including clarithromycin resistant *Mab*. Also, modified sequences by truncations of S5 (S52), S6 (S61, S62 and S63) and KLK (KLK1 and KLK2) were included.

**Table 3. *In vitro* interaction effects between AMPs (S61, S62, S63) and clarithromycin against *M. abscessus* clinical isolates.**

| Isolates | Type of CLA resistance | MIC of CLA (µg/ml) | MIC (µg/ml) | | FICI[a] | MIC (µg/ml) | | FICI[a] | MIC (µg/ml) | | FICI[a] |
|---|---|---|---|---|---|---|---|---|---|---|---|
| | | | S61 alone | Combined CLA (µg/ml) + S61 (µM) | | S62 alone | Combined CLA (µg/ml) + S62 (µM) | | S63 alone | Combined CLA (µg/ml) + S63 (µM) | |
| MAB01 | Acquired | 1,024 | 6.25 | 32/2.34 | 0.41 ±0.17 (Syn) | 12.5 | 96/2.34 | 0.28 ±0.13 (Syn) | 6.25 | 256/2.34 | 0.63 ±0.17 (Add) |
| MAB02 | Inducible | 64 | >400 | 64/6.25 | 1.02 ±0.00 (Ind) | >400 | 64/3.13 | 1.01 ±0.00 (Ind) | >400 | 64/14.06 | 1.04 ±0.04 (Ind) |
| MAB03 | Acquired | 8 | 6.25 | 2/0.20 | 0.28 ±0.00 (Syn) | 25 | 1.5/2.34 | 0.38 ±0.17 (Syn) | 12.5 | 4/0.20 | 0.52 ±0.00 (Add) |
| MAB05 | Inducible | 16 | >400 | 16/3.13 | 1.01 ±0.00 (Ind) | >400 | 16/3.13 | 1.01 ±0.00 (Ind) | >400 | 16/3.13 | 1.01 ±0.00 (Ind) |
| MAB07 | Inducible | 16 | >400 | 0.25/6.25 | 0.03 ±0.00 (Syn) | >400 | 0.25/3.13 | 0.02 ±0.00 (Syn) | >400 | 0.25/9.38 | 0.04 ±0.01 (Syn) |
| MAB09 | Inducible | 8 | >400 | 8/3.13 | 1.01 ±0.00 (Ind) | >400 | 6/3.13 | 0.76 ±0.35 (Add) | >400 | 4/6.25 | 0.52 ±0.00 (Add) |
| MAB10 | Acquired | 1,024 | >400 | 32/6.25 | 0.05 ±0.00 (Syn) | >400 | 32/18.75 | 0.08 ±0.02 (Syn) | >400 | 8/25 | 0.07 ±0.00 (Syn) |
| MAB11 | Inducible | 512 | >400 | 128/75 | 0.67 ±0.08 (Add) | >400 | 192/3.13 | 0.76 ±0.35 (Add) | >400 | 192/3.13 | 0.76 ±0.35 (Add) |
| MAB12 | Acquired | 1,024 | >400 | 1,024/3.13 | 1.01 ±0.00 (Ind) | >400 | 1,024/3.13 | 1.01 ±0.00 (Ind) | >400 | 1,024/3.13 | 1.01 ±0.00 (Ind) |
| MAB14 | Acquired | 512 | >400 | 4/37.5 | 0.10 ±0.04 (Syn) | >400 | 4/75 | 0.20 ±0.09 (Syn) | >400 | 32/50 | 0.19 ±0.00 (Syn) |

**Note:** AMP, antimicrobial peptide; MIC, minimal inhibitory concentration; CLA, clarithromycin; FICI, fractional inhibitory concentration index.

Ten clinical isolate representatives from the phylogenetic tree covering inducible and acquired resistances of two *Mab* subsp. that were selected for the AMP-clarithromycin interaction assay. The data exhibited mean ± S.D. of FICI values that were measured in two independent experiments.

[a]FICI interpretation: < 0.5: synergy (Syn); 0.5–1.0: additive (Add); > 1–4.0: indifference (Ind); > 4.0: antagonism (Ant).

Gray-shaded boxes show synergistic interaction.

Firstly, the activities of thirteen AMPs were screened against *Mab* ATCC19977 strain. Only the modified AMPs, including bovine myeloid analogs (S61, S62 and S63) and a flesh fly analog (KLK1), showed antimicrobial activity with MIC values ranging from 200–400 µg/ml. Compared to the parent AMP S6, the derivatives S61, S62, and S63 had no glycine (G), glycine-tyrosine (G-Y) and glycine-tyrosine-lysine (G-Y-K) residues. The parent KLK, derivatives KLK1 and KLK2 had no lysine (K) and leucine-lysine (L-K). Except for KLK2, these analogs had higher antimicrobial activity compared to their parent AMPs. The alteration of amino acid residues might increase the antimicrobial activity due to the higher hydrophobicity of their analogs that allowed better interaction with the pathogen cell surface [58]. Compared to other AMPs antimicrobial activity against other bacteria such as CRE had ranges of $MIC_{50}$ at 16->50 µM [45] and against *Acinetobacter baumannii* had ranges of MIC at 4–128 µg/ml reported [59]. The current study had MIC results for the majority of AMPs >400 µg/ml. This indicates that intrinsic antibiotic resistance of *Mab* is also highly resistant to naturally

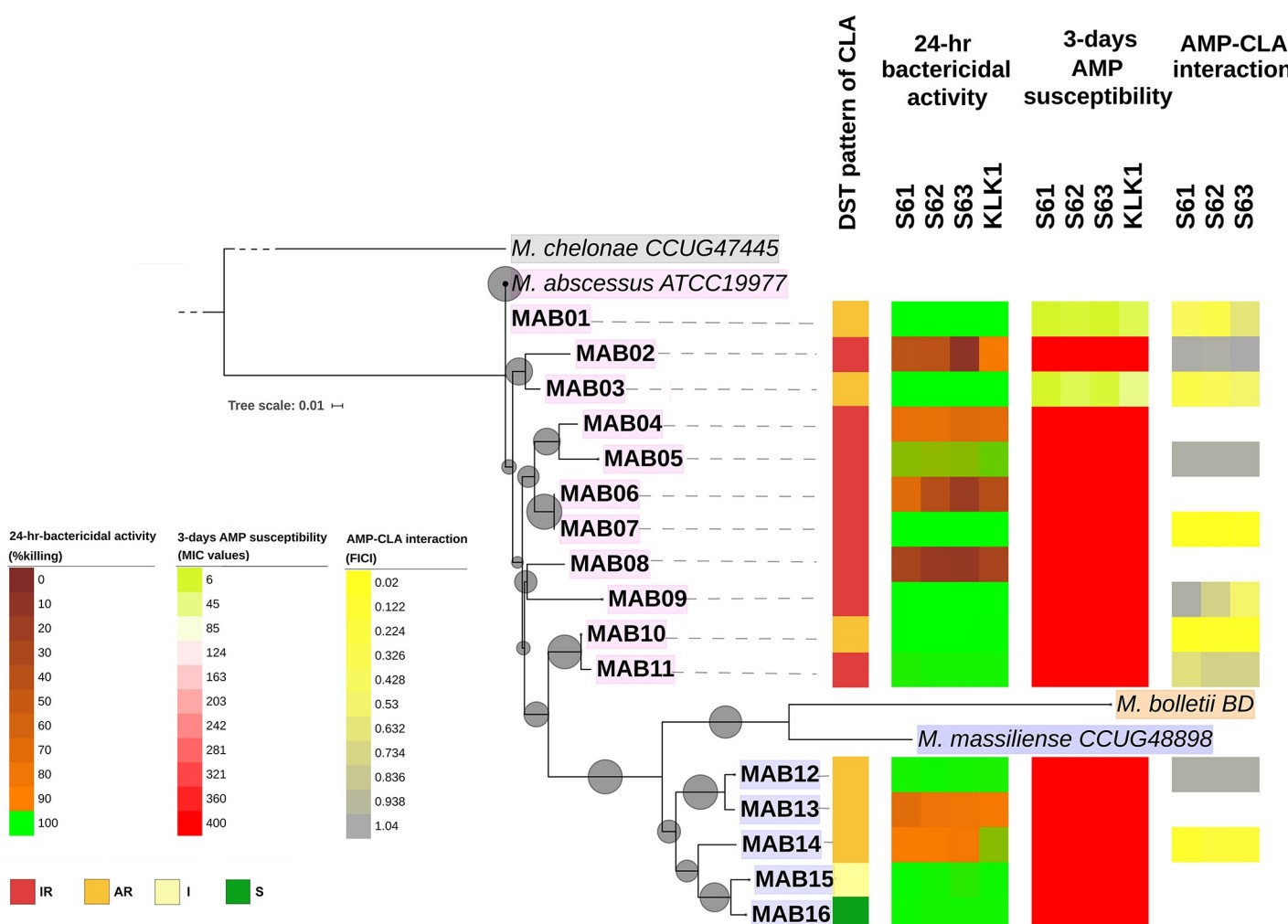

**Fig 3. The phylogeny of *M. abscessus* and clarithromycin/ AMPs susceptibility patterns.** A whole genome-based tree of 1,000 bootstraps from 3,180 SNPs is shown. The sequences of reference strains *M. chelonae* CCUG47445, *M. abscessus* subsp. *abscessus* ATCC19977, *M. abscessus* subsp. *bolletii* BD, and *M. abscessus* subsp. *massiliense* CCUG48898 were included without phenotypic results. Ten clinical isolates as representatives from the phylogenetic tree including inducible and acquired resistance of two *Mab* subspecies were selected for the AMP-clarithromycin interaction assay. AR, acquired resistance; CLA, clarithromycin; DST, drug susceptibility testing; FICI, fractional inhibitory concentration index; I, intermediate; IR, inducible resistance; MIC, minimum inhibitory concentration; S, susceptible.

occurring AMPs and only modified AMPs tended to increase their antimicrobial activities against clarithromycin resistant *Mab*.

All four potential AMPs (S61, S62, S63, and KLK1) were tested with sixteen clinical isolates of *Mab* selected based on the clarithromycin resistance and subspecies by determining the bactericidal activities within 24-hours. At 24 h, it was aimed to investigate the early antimicrobial activities of the potential AMPs. It was found that 62.5% of *Mab* showed a good bactericidal response to these AMPs at a low MIC level (50 μg/ml) within 24 h and 99% or more of *Mab* cells were killed based on the CFU assay. After further incubation for three days according to the standard DST for *Mab*, however, visible growths of *Mab* subpopulations were found. As a fresh medium for AMPs was not replaced in the assay, this might indicate that the stability of the AMPs is limited to only 24 h. By altering the culturing environment, some proteolytic enzymes might be degraded altering the antimicrobial activity [60, 61]. The limited performance of short-acting AMPs might be compensated for by sequential administration and/or

with the combination of antibiotics. Alternatively, only 12.5% of clarithromycin resistant *Mab* was susceptible with AMPs alone. This result indicates the nature of the high resistance properties of clarithromycin resistant *Mab*.

The toxicity of these four potential AMPs with hemolytic activity on human RBCs and viability of PBMCs was tested. It was found that all AMPs exhibited low hemolytic effects on RBCs and low PBMC deaths at concentrations ranging from 6.25–25 μg/ml that were lower or around their MIC levels. S63 showed the lowest toxicity. S61 also had comparable low toxicity compared to S63. Hence, these S6 analogs had a higher potential for clinical applications. KLK1 that had the highest hydrophobicity showed the highest hemolytic toxicity and lowest PBMC viability. Accordingly, increasing hydrophobicity may not only increase the antimicrobial activity but also the side effects to the host cells [58]. Hence, KLK1 was not included in later assays.

Clarithromycin is still the drug of choice for the treatment of *Mab* infection. Additional antibiotics such as intravenous amikacin plus either cefoxitin or imipenem may be added in the treatment regimen in case of clarithromycin resistance [62]. The additional antibiotics might still not be effective due to the increased side effects [63] and the high rate of treatment failure that still remains [64]. The antimicrobial activity of clarithromycin combined with these three AMP candidates using ten clarithromycin isolates as representatives from the phylogenetic tree were further tested. It was found that half of the clarithromycin resistant isolates had synergistic interactions. None of the antagonistic interactions of these AMPs and clarithromycin were found. With the synergistic effects, the average MICs of clarithromycin alone were largely reduced by 54-fold then combination treatments of each of three S6 analogs. Two *Mab* isolates with a high clarithromycin MIC at 1,024 μg/ml were killed by lower MICs at 8 and 32 μg/ml when treated with the AMP combinations. Also, these combinations effectively killed three isolates that had MIC values of AMPs greater than 400 μg/ml. The clarithromycin-AMP synergistic combination radically reduced the amount of AMPs required for the treatment, e.g. from 400 μg/ml to 3.13 μg/ml. This approach helps both treatment cost and toxicity reduction compared to treatment with an AMP alone. Thus, this study provides evidence to support that these novel potential AMPs might be used as an adjunct therapeutic approach in some clarithromycin resistant *Mab* infections. Regarding the needs for novel treatment options, treatment with a combination of AMPs might be effective in some cases with clarithromycin resistant *Mab* infections. As only half of the clarithromycin resistant *Mab* was susceptible to clarithromycin-AMP combination therapy, however, drug-AMP susceptibility tests might be needed before clinical application. Susceptibility testing for both AMPs and antibiotics might be required. Also, the clinical application of AMP for *Mab* infected patients is still unclear, such as the administrative approach and the half-lives of AMP *in vivo*. Further *in vivo* evaluation of the AMPs against *Mab* infections is needed.

The pattern of AMPs susceptibility and the phylogenetic tree or the clarithromycin resistant types (inducible and acquired resistance) were further investigated [8]. No clear-cut association between AMP susceptibility and clarithromycin resistance was found. The clarithromycin susceptible strains could resist AMPs. The AMPs-clarithromycin synergistic effect were found in both inducible and acquired clarithromycin strains, hence AMPs could be used for both resistant types. The colony morphotypes of *Mab* associated with biofilm formation and prolonged intracellular survival were reported [65]. With different cell wall surfaces biofilm formation might differ with AMP interaction. Here, the association between the colony morphotypes and AMP susceptibility were not observed. It was observed that none of inducible clarithromycin resistant isolates were susceptible to the 3-days AMP susceptibility test. Four out of 5 isolates showed synergistic activity in AMP-clarithromycin combinations that were acquired resistance and the fifth showed inducible resistance to clarithromycin. This

might indicate that the inducible resistant strains might be more highly resistant to AMPs compared to acquired resistant strains. A larger number of the tested strains allowing statistical analysis should be done to clarify such associations. In addition, the drug susceptibility test based on WGS analysis is still pending as the mutation database is not completed and the analysis take very long time to finish. Then, we have separated this objective out of the scope of this study.

The major limitation of this study was the limited number of the tested *Mab* isolates. This was difficult to test by statistical analysis. This is, so far, the largest number of clarithromycin resistant *Mab* strains tested with AMPs. *Mab* is a prolonged intracellular pathogen with varied ability to produce biofilm, biofilm-forming smooth morphotypes and non-biofilm forming rough morphotypes [65]. Although various *Mab* isolates of both rough and smooth morphotypes were included in these experiments, DST were not determined in the biofilm-producing state of *Mab*. These experiments were only under *in vitro* conditions; the *in vivo* response of AMPs and CLA could be varied depending on the host environment. Additional studies that investigate the in-depth assessment with a larger number of isolates, including the biofilm-producing state and *in vivo* experiments are likely warranted.

In conclusion, the antimicrobial activities of AMPs against clarithromycin resistant *Mab* were assessed. Only AMPs with truncated modifications showed antimicrobial activity against clarithromycin resistant *Mab*. Three novel AMPs, S61, S62, and S63, based on S6 truncated modifications exhibited antimicrobial activity against more than half of clarithromycin resistant *Mab*. Variable antimicrobial activities of AMPs against clarithromycin resistant *Mab* were found but no associations between AMP susceptibility and phylogeny or clarithromycin resistant types were found. Half of the clarithromycin resistant isolates provided synergistic interactions between clarithromycin and AMPs. The variable AMP susceptibilities of clarithromycin resistant *Mab* were demonstrated.

## Supporting information

**S1 Table. The raw data of 24-hour bactericidal activities of four AMPs against sixteen clinical isolates of *M*. *abscessus*.**
(XLSX)

**S2 Table. Summary result of toxic effect of human blood cells after treated with four potential peptides for 1 hour.**
(XLSX)

## Acknowledgments

We would like to acknowledge Emeritus Professor James A Will, University of Wisconsin-Madison for editing the MS via Publication Clinic KKU, Thailand.

## Author Contributions

**Conceptualization:** Kiatichai Faksri.

**Data curation:** Phantitra Sudadech.

**Formal analysis:** Phantitra Sudadech.

**Funding acquisition:** Kiatichai Faksri.

**Investigation:** Phantitra Sudadech, Orawee Kaewprasert.

**Methodology:** Kiatichai Faksri.

**Project administration:** Kiatichai Faksri.

**Resources:** Sittiruk Roytrakul, Ploenchan Chetchotisakd, Kiatichai Faksri.

**Supervision:** Sittiruk Roytrakul, Ploenchan Chetchotisakd, Sakawrat Kanthawong, Kiatichai Faksri.

**Validation:** Phantitra Sudadech, Kiatichai Faksri.

**Visualization:** Phantitra Sudadech, Kiatichai Faksri.

**Writing – original draft:** Phantitra Sudadech, Kiatichai Faksri.

**Writing – review & editing:** Auttawit Sirichoat, Kiatichai Faksri.

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
