## [Decision Letter · Decision Letter 0]

26 Jul 2021

PONE-D-21-19959

Assessment of in vitro activities of novel modified antimicrobial peptides against clarithromycin resistant Mycobacterium abscessus

PLOS ONE

Dear Dr. Faksri,

Thank you for submitting your manuscript to PLOS ONE. After careful consideration, we feel that it has merit but does not fully meet PLOS ONE’s publication criteria as it currently stands. Therefore, we invite you to submit a revised version of the manuscript that addresses the points raised during the review process.

Please address all reviewer comments point by point and revise the manuscript.

We look forward to receiving your revised manuscript.

Kind regards,

Iddya Karunasagar

Academic Editor

PLOS ONE

Journal Requirements:

2. Please provide additional details regarding participant consent to collect blood samples for isolation of RBC and PBMCs. In the Methods section, please ensure that you have specified how consent was obtained or whether the research ethics committee or IRB specifically waived the need for their consent.

5. Thank you for stating the following in the Funding Section of your manuscript:

“This study was supported by the Invitation Research, Faculty of Medicine, Khon Kaen University (Grant number: IN62307) and Research and Diagnostic Center for Emerging Infectious Diseases (RCEID), Khon Kaen University, Khon Kaen, Thailand.”

We note that you have provided additional information within the Funding Section that is not currently declared in your Funding Statement. Please note that funding information should not appear in the Acknowledgments section or other areas of your manuscript. We will only publish funding information present in the Funding Statement section of the online submission form.

Additional Editor Comments:

The reviewer has raised number of questions for which clarifications/explanations are needed. Please revise considering all reviewer comments point by point.

Reviewers' comments:

Reviewer's Responses to Questions

**Comments to the Author**

1. Is the manuscript technically sound, and do the data support the conclusions?

Reviewer #1: Partly

2. Has the statistical analysis been performed appropriately and rigorously? 

Reviewer #1: Yes

3. Have the authors made all data underlying the findings in their manuscript fully available?

Reviewer #1: Yes

4. Is the manuscript presented in an intelligible fashion and written in standard English?

Reviewer #1: No

5. Review Comments to the Author

Reviewer #1: Major comments– M abscessus does exists as biofilms during infections and the resistance pattern of biofilm vs broth grown Mab are highly varied. Why have the authors restricted to the broth form of MAB. Line 312 talks about anti-biofilm effect of AMPs on Streptococcus. The in vitro synergism and anti-microbial effect may be very different from the clinical scenario unless screened on a biofilm model and warrants further investigation before moving forward. Discussion needs to throw light on specific aspects like treatment response and duration due to use of AMP. Please highlight other drug resistance if any identified through sequencing and any similarities and differences in mutation patterns among cluster of inducible or acquired resistant strains.

Line 51 – Mention the duration of treatment

Line 58 – Please mention which other bacteria and if any mycobacterial species were tested with AMPs

Line 62 – “Polydim-I reduced 40 to 50% of Mab subsp. massiliense infected macrophage”. This sentence refers to mycobacteria or macrophages. It is confusing.

Table2 shows MAB13 as acquired while it is increasing from day 3 to day 14 with respect to clarithromycin.

Several typos and grammar improvement for discussion recommended – line 373 (regarding); 378 – patients and administration etc.,

6. PLOS authors have the option to publish the peer review history of their article (what does this mean?). If published, this will include your full peer review and any attached files.

Reviewer #1: No

---

## [Author Response · Author response to Decision Letter 0]

13 Aug 2021

Response to reviewer comments

Thank you very much for your comments and suggestions. We would like to answer your comments and suggestions as follows:

Journal Requirements:

Answer: The manuscript formatting has been checked and the format revised accordingly. We hope that the revised version of the manuscript is according to PLOS ONE's style requirements.

2. Please provide additional details regarding participant consent to collect blood samples for isolation of RBC and PBMCs. In the Methods section, please ensure that you have specified how consent was obtained or whether the research ethics committee or IRB specifically waived the need for their consent.

Answer: A sentence has been added to the Materials and methods section to state this information, as follows:

(Lines 93-96) Informed consent was not required for this study. All specimens including isolates and blood samples were obtained from routine practice in which patient's information were deidentified. The study protocol was approved by the Khon Kaen University Ethics Committee for Human Research (HE611496). The document in which the IRB specifically waived the need for their consent is contained in the supplemental file.

Answer: English usage of the final version of this manuscript was reviewed by Emeritus Professor James A Will, University of Wisconsin-Madison for editing the MS via Publication Clinic KKU, Thailand (jawenator@gmail.com). He is the senior editor for the Faculty of Medicine and successfully edits 80-100 medical manuscripts for this Faculty a year. We will not make any changes after his editing without his further editing. The manuscript has further been revised for typos and grammatical errors. We hope this new version meets the high standards of PLOS ONE.

Answer: We added the funding on online submission form, as follows:

This study was supported by the Invitation Research, Faculty of Medicine, Khon Kaen University (Grant number: IN62307) and National Research Council of Thailand (Grant number: NRC MHESI 483/2563). The funders had no role in study design, data collection and analysis, decision to publish, or preparation of the manuscript.

5. Thank you for stating the following in the Funding Section of your manuscript:

“This study was supported by the Invitation Research, Faculty of Medicine, Khon Kaen University (Grant number: IN62307) and Research and Diagnostic Center for Emerging Infectious Diseases (RCEID), Khon Kaen University, Khon Kaen, Thailand.”

We note that you have provided additional information within the Funding Section that is not currently declared in your Funding Statement. Please note that funding information should not appear in the Acknowledgments section or other areas of your manuscript. We will only publish funding information present in the Funding Statement section of the online submission form.

Answer: We removed funding-related text and the information of competing interests from the manuscript and these sentences are stated in the online submission form.

 

Reviewer #1:

Q#1 M abscessus does exists as biofilms during infections and the resistance pattern of biofilm vs broth grown Mab are highly varied. Why have the authors restricted to the broth form of MAB.

Answer: The broth state of the experiment is the easiest way to control environmental condition of the experiment and drug exposure to the pathogens. The point that the experimental condition does not cover the biofilm producing state is the limitation of our study. We have added this limitation in the last paragraph of the Discussion section as follows:

(Lines 409-417) Mab is a prolonged intracellular pathogen with varied ability to produce biofilm, biofilm-forming smooth morphotypes and non-biofilm forming rough morphotypes [63]. Although various Mab isolates of both rough and smooth morphotypes were included in these experiments, DST were not determined in the biofilm producing state of Mab. DST were not determined in the biofilm producing state of Mab. These experiments were only under in vitro conditions; the in vivo response of AMPs and CLA could be varied depending on the host environment. Additional studies that investigate the in-depth assessment with a larger number of isolates, including the biofilm producing state and an in vivo experiment are likely warranted.

Q#2 Line 312 talks about anti-biofilm effect of AMPs on Streptococcus. The in vitro synergism and anti-microbial effect may be very different from the clinical scenario unless screened on a biofilm model and warrants further investigation before moving forward. Discussion needs to throw light on specific aspects like treatment response and duration due to use of AMP.

Answer: Mab is a prolonged intracellular pathogen with varied ability to produce biofilm, biofilm-forming smooth morphotypes and non-biofilm forming rough morphotypes [63]. Although various Mab isolates of both rough and smooth morphotypes were included in these experiments, DST were not determined in the biofilm producing state of Mab. DST were not determined in the biofilm producing state of Mab. This is the limitation of our study. We stated this as the limitation in the discussion (Lines 409-413).

Q#3 Please highlight other drug resistance if any identified through sequencing and any similarities and differences in mutation patterns among cluster of inducible or acquired resistant strains.

Answer: The isolates that we included are the CLA-resistant Mab isolates. The susceptibility test for amikacin which is another important drug for treatment of Mab infection is also available. We have included the susceptibility results of the additional drug in the Table 2. The drug susceptibility test based on WGS analysis is still pending as the mutation database is not compete and the analysis take very long time to finish. Then we have separated this objective out of the scope of this study.

Q#4 Line 51 – Mention the duration of treatment

Answer: A new text has been added to the Introduction section to state, as follows:

(Lines 51-52) The duration of treatment for Mab infection depends on the clinical syndrome and lasts from 4 weeks to 12 months.

Q#5 Line 58 – Please mention which other bacteria and if any mycobacterial species were tested with AMPs

Answer: In this study, no other Mycobacterium species was tested but references are provided to our work. A new text has been added to the Introduction section to state and the references have been added, as follows:

(Lines 61-64) Several research teams have reported AMPs activity against Mycobacterium tuberculosis [10-27] and other NTMs such as Mycobacterium avium [12, 19, 28, 29], Mycobacterium smegmatis [12, 20, 30], Mycobacterium vaccae [31], Mycobacterium bovis [32] and Mycobacterium marinum [33].

Q#6 Line 62 – “Polydim-I reduced 40 to 50% of Mab subsp. massiliense infected macrophage”. This sentence refers to mycobacteria or macrophages. It is confusing.

Answer: This sentence refers to macrophages. We have modified this sentence as follows:

(Lines 67-69) Polydim-I treatment of macrophages infected with different Mab subsp. massiliense strains reduced the bacterial load by 40 to 50% [35].

Q#7 Table2 shows MAB13 as acquired while it is increasing from day 3 to day 14 with respect to clarithromycin.

Answer: Inducible resistance was inferred by changes in MIC values from “susceptible” at day 3 to “resistant” at day 14. Isolates that were resistant on day 3 and thereafter were regarded as demonstrating acquired resistance. Therefore, MAB13 had resistance from day 3 to day 14, which was defined as acquired resistance.

Q#8 Several typos and grammar improvement for discussion recommended – line 373 (regarding); 378 – patients and administration etc.,

Answer: English usage of the final version of this manuscript will be reviewed by Emeritus Professor James A Will, University of Wisconsin-Madison for editing the MS via Publication Clinic KKU, Thailand (jawenator@gmail.com). The current revised version of the manuscript has further been revised for typos and grammatical errors. We hope this new version meets the high standards of PLOS ONE.

---

## [Decision Letter · Decision Letter 1]

25 Oct 2021

PONE-D-21-19959R1Assessment of in vitro activities of novel modified antimicrobial peptides against clarithromycin resistant Mycobacterium abscessusPLOS ONE

Dear Dr. Faksri,

Thank you for submitting your manuscript to PLOS ONE. After careful consideration, we feel that it has merit but does not fully meet PLOS ONE’s publication criteria as it currently stands. Therefore, we invite you to submit a revised version of the manuscript that addresses the points raised during the review process.

We look forward to receiving your revised manuscript.

Kind regards,

Iddya Karunasagar

Academic Editor

PLOS ONE

Journal Requirements:

Additional Editor Comments:

Please address minor comments of the reviewer

Reviewers' comments:

Reviewer's Responses to Questions

**Comments to the Author**

1. If the authors have adequately addressed your comments raised in a previous round of review and you feel that this manuscript is now acceptable for publication, you may indicate that here to bypass the “Comments to the Author” section, enter your conflict of interest statement in the “Confidential to Editor” section, and submit your "Accept" recommendation.

Reviewer #1: All comments have been addressed

2. Is the manuscript technically sound, and do the data support the conclusions?

Reviewer #1: Partly

3. Has the statistical analysis been performed appropriately and rigorously? 

Reviewer #1: Yes

4. Have the authors made all data underlying the findings in their manuscript fully available?

Reviewer #1: Yes

5. Is the manuscript presented in an intelligible fashion and written in standard English?

Reviewer #1: No

6. Review Comments to the Author

Reviewer #1: 1. Please add "The drug susceptibility test based on WGS analysis is still pending

as the mutation database is not compete and the analysis take very long time to finish.

Then we have separated this objective out of the scope of this study" to the main text.

2. Please add reference to "The duration of treatment for Mab infection depends on the clinical

syndrome and lasts from 4 weeks to 12 months" in line 51

3. Spell check to be rigorously done

7. PLOS authors have the option to publish the peer review history of their article (what does this mean?). If published, this will include your full peer review and any attached files.

Reviewer #1: No

---

## [Author Response · Author response to Decision Letter 1]

27 Oct 2021

Response to reviewer comments

Thank you very much for your comments and suggestions. We would like to answer your comments and suggestions as follows:

Journal Requirements:

Answer: Thank you very much for your suggestions. We checked the references have been corrected and completed. The reference list has been rearranged because two references have been added according to the reviewer suggested as follows:

(Lines 450-455) 

6. Strnad L, Winthrop KL. Treatment of Mycobacterium abscessus Complex. Semin Respir Crit Care Med. 2018;39(3):362-76. https://doi.org/10.1055/s-0038-1651494 PMID: 30071551

7. Weng YW, Huang CK, Sy CL, Wu KS, Tsai HC, Lee SS. Treatment for Mycobacterium abscessus complex-lung disease. J Formos Med Assoc. 2020;119 Suppl 1:S58-S66. https://doi.org/10.1016/j.jfma.2020.05.028 PMID: 32527504

 

Comments to the Author

Review Comments to the Author

Reviewer #1:

1. Please add "The drug susceptibility test based on WGS analysis is still pending as the mutation database is not compete and the analysis take very long time to finish. Then we have separated this objective out of the scope of this study" to the main text.

Answer: We have added these sentences as the limitation in the Discussion section as follows:

 “The drug susceptibility test based on WGS analysis is still pending as the mutation database is not completed and the analysis take very long time to finish. Then, we have separated this objective out of the scope of this study.” (Lines 405-407).

2. Please add reference to "The duration of treatment for Mab infection depends on the clinical syndrome and lasts from 4 weeks to 12 months" in line 51

Answer: The reference has been added, as follows:

The duration of treatment for Mab infection depends on the clinical syndrome and lasts from 4 weeks to 12 months [6, 7]. (Lines 52-53)

6. Strnad L, Winthrop KL. Treatment of Mycobacterium abscessus Complex. Semin Respir Crit Care Med. 2018;39(3):362-76. https://doi.org/10.1055/s-0038-1651494 PMID: 30071551

7. Weng YW, Huang CK, Sy CL, Wu KS, Tsai HC, Lee SS. Treatment for Mycobacterium abscessus complex-lung disease. J Formos Med Assoc. 2020;119 Suppl 1:S58-S66. https://doi.org/10.1016/j.jfma.2020.05.028 PMID: 32527504

3. Spell check to be rigorously done

Answer: Thank you very much for your suggestions. Before submission, the final version of this revised manuscript was reviewed by Emeritus Professor James A Will, University of Wisconsin-Madison for editing the MS via Publication Clinic KKU, Thailand. The spelling check have been done. All typographical errors have been corrected. 

---

## [Decision Letter · Decision Letter 2]

2 Nov 2021

Assessment of in vitro activities of novel modified antimicrobial peptides against clarithromycin resistant Mycobacterium abscessus

PONE-D-21-19959R2

Dear Dr. Faksri,

We’re pleased to inform you that your manuscript has been judged scientifically suitable for publication and will be formally accepted for publication once it meets all outstanding technical requirements.

Kind regards,

Iddya Karunasagar

Academic Editor

PLOS ONE

Additional Editor Comments (optional):

All reviewer comments have been addressed satisfactorily.

Reviewers' comments:

Reviewer's Responses to Questions

**Comments to the Author**

1. If the authors have adequately addressed your comments raised in a previous round of review and you feel that this manuscript is now acceptable for publication, you may indicate that here to bypass the “Comments to the Author” section, enter your conflict of interest statement in the “Confidential to Editor” section, and submit your "Accept" recommendation.

Reviewer #1: All comments have been addressed

2. Is the manuscript technically sound, and do the data support the conclusions?

Reviewer #1: Partly

3. Has the statistical analysis been performed appropriately and rigorously? 

Reviewer #1: Yes

4. Have the authors made all data underlying the findings in their manuscript fully available?

Reviewer #1: Yes

5. Is the manuscript presented in an intelligible fashion and written in standard English?

Reviewer #1: Yes

6. Review Comments to the Author

Reviewer #1: The manuscript has addressed major concerns raised and most of the typographical errors have been corrected

7. PLOS authors have the option to publish the peer review history of their article (what does this mean?). If published, this will include your full peer review and any attached files.

Reviewer #1: No

---

## [Editor Report · Acceptance letter]

4 Nov 2021

PONE-D-21-19959R2 

Assessment of *in vitro* activities of novel modified antimicrobial peptides against clarithromycin resistant *Mycobacterium abscessus*

Dear Dr. Faksri:

I'm pleased to inform you that your manuscript has been deemed suitable for publication in PLOS ONE. Congratulations! Your manuscript is now with our production department. 

Kind regards, 

on behalf of

Dr. Iddya Karunasagar 

Academic Editor

PLOS ONE